# Impact of Social and Economic Determinants on the Prevalence of Childhood Overweight and Obesity: A Cross-Sectional Study from the ENPIV in Valencia, Spain

**DOI:** 10.3390/nu17122006

**Published:** 2025-06-15

**Authors:** Luis Cabañas-Alite, María Alonso-Asensi, Elena Rocher-Vicedo, Lidia Garcia-Garcia, Ruth Garcia-Barajas, Jose M. Martin-Moreno

**Affiliations:** 1Faculty of Health Sciences, European University of Valencia, Paseo de la Alameda 7, 46010 Valencia, Spain; lcabanas@uemc.es; 2Project Department, Official College of Dietitians-Nutritionists of the Valencian Community Region (CODiNuCoVa), Clariano St. 34, 46021 Valencia, Spain; 3Valencian Community Foundation for Strategic Promotion, Development and Urban Innovation, Joan Verdeguer St. 16, 46002 Valencia, Spain; 4Department of Preventive Medicine and Public Health, Faculty of Medicine, University of Valencia, Blasco Ibañez Avenue 15, 46010 Valencia, Spain

**Keywords:** malnutrition, socioeconomic status, food insecurity, childhood obesity, obesity risk factors

## Abstract

**Objective**: To characterize the nutritional status of the child population and to identify the most relevant determinants that could influence the early development of obesity and overweight. **Methods**: A descriptive cross-sectional study was conducted, recruiting a representative sample (698 schoolchildren, aged between 5 and 14, and obtaining information from a total of 414 households), using of anthropometric methods for nutritional assessment, the FIES scale, the KidMed index, and questions from the COSI survey. **Results**: Overall, 27% of the sample had healthy weight, 22.6% were overweight, and 18.1% were obese, with a higher prevalence among boys, and 86.7% of the sample did not adhere to dietary habits consistent with the Mediterranean diet. Food insecurity was present in 45.6% of the sample. A mean weight increase of 2.6 kg (95% CI: 1.0–4.3) was observed for each additional kilogram the child weighed at birth, 2.0 kg (95% CI: −0.2–4.3) in children living in households with some difficulty making ends meet, and 4.8 kg (95% CI: 1.3–8.3) in those from households reporting that they do not or barely manage to make ends meet. A statistically significant association was found with gross annual household income, with children from households earning less than EUR 12,000 having a 3.6 times higher risk of overweight/obesity compared to those from households earning more than EUR 36,000. **Conclusions**: The prevalence of obesity and overweight is considerably higher in low-income families and is significantly associated with family economic capacity. Continued epidemiological surveillance of these conditions and their relationship with social determinants is necessary.

## 1. Introduction

For decades, excess body fat in the form of obesity and overweight has been a serious issue within the pediatric population, significantly increasing the risk of developing other non-communicable diseases, both in childhood and later in adulthood. This condition has been described as the global pandemic of this millennium, requiring a transdisciplinary approach and urgent action [1,2].

The rising prevalence of this condition led all global regions to research its extent and the underlying factors contributing to its increase worldwide. In Europe, the *Childhood Obesity Surveillance Initiative* (COSI) was established in 2007 in response to the need to standardize data on this issue across the region. For its fifth edition, 45 countries participated, assessing their child populations between 2018 and 2020, with the results published in November 2022 [3]. The findings ranked Spain as the fourth country in terms of having the highest percentage of childhood overweight and obesity among boys (behind Cyprus, Greece, and Italy) and second among girls (behind Cyprus). This study reported an average prevalence of childhood overweight and obesity at 29% for children aged 7 to 9 years across the 33 participating countries, with highly variable prevalence rates between nations (for instance, 6% overweight in boys in Tajikistan, compared to 43% in Cyprus, with Germany falling in the middle at 18%) [3]. These findings echoed those of the previous edition, which already highlighted economic inequities as a key factor explaining the higher rates of childhood obesity and overweight in certain regions [4].

To monitor the prevalence of childhood obesity and participate in COSI, Spain has relied on the Monitoring of Diet, Physical Activity, Child Development, and Obesity in Spain study (ALADINO, which stands for *Alimentación*, *Actividad Física*, *Desarrollo Infantil y Obesidad*) since 2011. Its latest report from 2019, which included 16,665 schoolchildren from 276 primary schools, indicated a stabilization in weight-related issues since 2015, with 41.3% of children classified as overweight (23.2% overweight and 18.1% obese), and a significant gap associated with household income levels pointing to an ongoing risk of increasing trends in obesity and overweight [5]. Similar initiatives in Spain include the AVENA study (*Alimentación y Valoración del Estado Nutricional de los Adolescentes study*), which identified a 36.3% prevalence of overweight and obesity among adolescents in 2005 [6]. Additionally, in many countries, private or non-institutional research initiatives have emerged, such as the PASOS study (Physical Activity, Sedentarism, lifestyles and Obesity in Spanish Youth Study) by the Gasol Foundation, which assessed a total of 2892 children aged 8 to 16 in its 2019 and 2022 editions. This study estimated the prevalence of obesity and overweight at 33.4% [7].

Among the participants in the ALADINO 2019 study [5] (*ALimentación*, *Actividad física*, *Desarrollo INfantil y Obesidad study*), household socioeconomic conditions, lifestyle habits, and the surrounding environment were examined. The conclusion was that childhood obesity was more prevalent among schoolchildren with low consumption of fruit and vegetables, high intake of sweets, low levels of time spent on physical activity, and high levels of screen time. These factors were associated with low household incomes, such that childhood obesity was significantly more frequent in schools located in districts with high rates of child poverty, with obesity being seven times more common among girls (18.2% compared to 2.5%) and twice as high among boys (21.7% compared to 10.7%) [8].

These findings are consistent with the European context. The latest European study assessing the prevalence of childhood overweight—obesity (collecting data from 123,487 children across 24 European countries) found an inverse relationship between family purchasing power and childhood overweight or obesity, especially in wealthier countries such as Czechia (OR: 1.59 95% CI: 1.00–2.54), Germany (OR: 1.57 95% CI: 0.95–2.60), France (OR: 1.78 95% CI: 1.36–2.32), and Ireland (OR: 1.56 95% CI: 1.00–2.43). This relationship was less pronounced in countries with the lowest GDPs in Europe, such as Albania (OR: 0.46 95% CI: 0.34–0.62), Georgia (OR: 0.77 95% CI: 0.60–0.98), and Romania (OR: 0.79 95% CI: 0.64–0.97)] [9].

There is emerging evidence suggesting that poorer dietary and lifestyle habits may be linked to the experience of food insecurity within the household. To assess this issue, the Food Insecurity Experience Scale (FIES) was developed in 2013 [10] and incorporated into food security surveys in 2014 [11]. The scale does not measure food consumption or provide information on dietary quality; rather, its aim is to quantify food insecurity based on access to food in order to explore health determinants that might contribute to such insecurity. Since 2017, the FIES has been adopted as the indicator for Sustainable Development Goal 2 [12].

Given this context, the objective of this study was to characterize the nutritional status of the child population in a healthcare region of the city of Valencia (Spain) and to identify various social, economic, and dietary determinants that could influence the early development of obesity and overweight.

## 2. Materials and Methods

### 2.1. Study Design and Sampling

This descriptive, cross-sectional study was conducted in the Health Department–Valencia–Dr Peset (serving a population of approximately 300,000 people). Schoolchildren aged 5 to 14 years were recruited from their schools, with information collected from their parents or legal guardians.

The sample was representative of the population studied, utilising a cluster sampling design in which schools served as the primary sampling units. Recruitment was universal, with oversampling implemented to account for a 5% margin of error at a 95% confidence level, considering the total population of each participating school. The recruitment period spanned two academic years (2021–2022 and 2022–2023), with data collection taking place over one academic year (2022–2023). STROBE cross-sectional reporting guidelines [13] were adopted, and responses to each question can be found in the Appendix A).

The inclusion criteria were (a) schoolchildren aged 5 to 14 years and (b) those with informed consent provided by their parents or legal guardians. The exclusion criteria included (a) children under 5 or over 14 years of age; (b) pregnant girls or adolescents; (c) schoolchildren diagnosed with any conditions that could affect body composition, such as type 1 diabetes, congenital metabolic disorders, cystic fibrosis, neoplastic diseases, inflammatory bowel disease, nephropathies, cardiopathies, coeliac disease, or those undergoing pharmacological treatments that may alter body composition, weight, or height (e.g., growth hormone); and (d) individuals who withdrew informed consent at any point during the study.

### 2.2. Study Measurements

The measurements were carried out by trained personnel using calibrated equipment, and the same staff conducted follow-up phone calls to households to survey variables related to social determinants, diet, and food security.

#### 2.2.1. Household Food Insecurity Status and Dietary Habits

Food security status during the past 12 months was assessed using the eight-question Food Insecurity Experience Scale (FIES), Spanish language version, developed by UN FAO. Questions addressed issues ranging from difficulties in accessing food due to lack of money or other resources to going without eating for an entire day. Households with a raw score of zero were categorized as ‘food secure’; those with raw scores of 1–3 as ‘mild food insecurity’; and those with scores of 4–8 as ‘moderate–severe food insecurity’.

To assess the dietary habits of the child population, the KidMed questionnaire (v. 2004) was employed, with the following categories based on the overall score: ≤3, indicating a very low-quality diet; 4 to 7, a need for improvement in dietary patterns; and ≥8, indicating an optimal Mediterranean diet [14]. This tool was validated for this population [14]. Responses to each question from these surveys can be found in the Appendix A.

#### 2.2.2. Anthropometric Indicators of Nutritional Status

The anthropometric variables were measured following the ISAK (*International Society for the Advancement of Kinanthropometry*) cineanthropometric method as a technique to assess and analyze the body composition of the individuals studied. The variables included were as follows: weight (kg, SECA 803 portable scale, SECA GmbH & Co., KG, Hamburg, Germany; measurement resolution ± 100 g), height (cm, SECA 213 portable stadiometer, SECA GmbH & Co., KG, Hamburg, Germany; measurement resolution ± 1 mm), triceps skinfold (mm, Innovare3 CESCORF^®^ calliper, CESCORF Equipamentos para Esporte Ltda., Porto Alegre, Brazil; measurement resolution ± 1 mm), and the following girth measurements (in cm, CESCORF^®^ flexible, inelastic steel tape, CESCORF Equipamentos para Esporte Ltda., Porto Alegre, Brazil; measurement resolution ± 1 mm, 6 mm wide): upper arm, waist, and hip.

From the above data, the following were calculated: Z-scores (weight-for-age, height-for-age, weight-for-height, BMI-for-age), upper arm muscle area, waist-to-hip ratio, and waist-to-height ratio. The population was classified according to their nutritional status based on the WHO growth reference charts for BMI-for-age [15].

#### 2.2.3. Covariates

Households were also surveyed regarding socioeconomic and educational health determinants for parents or legal guardians (including age, gender, level of education, current occupation, employment status, and approximate annual household income). In addition, parents or legal guardians were also asked about children’s lifestyle factors (such as screen time, and physical activity in minutes per day), and personal factors during early life (total breastfeeding in months, adherence to exclusive breastfeeding for six months, and birth weight). To obtain this information, we used the questions used by the COSI survey (questions 5 to 19), since the use of these questions has been validated for several European countries [16].

### 2.3. Data Analysis

To examine the relationship between two quantitative variables, the Pearson correlation coefficient, the linear regression slope (along with its 95% confidence interval), and the associated *p*-value for the non-nullity of the correlation coefficient (equivalent to the non-nullity of the slope of the fitted model) were estimated. To assess the relationship between a quantitative variable and a qualitative variable, the mean values of the quantitative variable observed in each group defined by the qualitative variable were compared, using independent samples Student’s *t*-tests for qualitative variables with two categories and ANOVA tests for those with more than two categories, applying the Bonferroni correction to assess significance between the groups for each category. Finally, a multivariate linear regression model was constructed for each variable related to nutritional status to evaluate the relationship between health variables and social determinants, assessing the goodness-of-fit using the estimated coefficient of determination for the final model. In all cases, the applicability conditions for the different statistical methods were verified. For sample descriptions, the means, percentiles, and standard deviations of the quantitative variables were calculated.

All analyses were performed using IBM SPSS Statistics v22.0 and R 4.2.1, considering two-tailed tests and a significance level of 5% throughout.

### 2.4. Ethics

This study was conducted according to the guidelines laid down in the Declaration of Helsinki and all procedures involving research study participants were approved by the Clinical Research Ethics Committee of the *Hospital Universitario Doctor Peset* of Valencia by means of an authorization report with CEIm Code 107.21. Written informed consent was obtained from all subjects.

## 3. Results

The final sample included 698 participants aged 5 to 14 years, from whom data were subsequently obtained for 414 households. The data related to nutritional status, gender, and age for both samples can be seen in Table 1, broken down by variable, with mean, median, and basic descriptive statistics, including the 95% confidence interval for the prevalence of different weight status categories in the total sample.

Regarding the prevalence of overweight and obesity, 398 schoolchildren (57%) had normal weight, while 22.6% were overweight, and 18.1% were obese. These figures varied by gender, with obesity being more frequent than overweight among boys (21.11% and 20.83%, respectively), while girls showed 14.8% obesity and 24.56% overweight. Overall, 40.7% of the sample was classified as having excess weight, with slight differences by gender (41.94% in boys and 39.36% in girls) for a similar sample size. The other body composition parameters and age ranges were comparable between genders. In the comparison between the total sample and the subset with sociodemographic variables, a slight increase in age was observed, along with age-related variables (height and muscular strength), without further significant differences.

### 3.1. Socio-Demographic Characteristics, Household Food Insecurity, and Nutritional Status

In Table 2, the personal characteristics at birth, dietary habits, and physical activity or screen time are presented by gender and weight status, based on the interviews conducted with the households in the sample (n = 414). Noteworthy is the low adherence to exclusive breastfeeding for six months (39.6% of the sample), higher birth weight in children with higher weight status as they age, and the fact that 86.7% of the sample did not follow dietary habits compatible with the Mediterranean diet. Regarding the FIES scale assessment, 45.7% of the sample experienced food insecurity (39.6% mild, and 6% moderate or severe).

In Table 3, the relationship between nutritional status and household income levels is presented for those who provided this information (22.20% did not respond to this question, leaving a sample of 322 households). The impact of income level on students’ nutritional status is evident, with observed improvements as household income increases. This relationship is significant for the variables BMI (*p*-value: <0.001), Z-score (*p*-value: 0.003), triceps skinfold thickness (*p*-value: 0.001), which relates to excess body fat, and waist (*p*-value: 0.002) and arm circumferences (*p*-value: 0.001). The same table further includes data on financial difficulties in reaching the end of the month, which further underscores the significant relationships between nutritional status and these variables, including weight (*p*-value: 0.013), Z-score (*p*-value: 0.001), BMI (*p*-value: <0.001), and waist (*p*-value: 0.008), hip (*p*-value: 0.006), and arm circumferences (*p*-value: 0.002), with a greater response rate (n = 401).

### 3.2. Multivariate Relationship of Nutritional Status with Health and Socio-Economic Determinants

In Table 4, multivariate linear regression models for nutritional status in relation to the analyzed determinants are presented. Two final models are shown. First, two factors were significantly linked with the students’ weight: birth weight (*p*-value: 0.002) and household’s ease in making ends meet (*p*-value: 0.016). There was an average increase in student weight of 2.6 kg (95% CI: 1.0–4.3) for each additional kilogram at birth, 2.0 kg (95% CI: −0.2–4.3) for students from households that have some difficulty making ends meet, and 4.8 kg (95% CI: 1.3–8.3) for those from households reporting they do not, or barely, manage to make ends meet. The remaining factors ceased to be significant in the presence of these in the final model.

Secondly, the same analysis was conducted for the Z-score, where the final model included birth weight (*p*-value: 0.005), household difficulty in making ends meet (*p*-value: <0.001), and annual household income (*p*-value: 0.009) as significant factors. Once these factors were included, other variables ceased to be significant. There was a statistically significant relationship with gross annual household income, with a 3.6-fold higher risk of overweight/obesity in students from households earning less than EUR 12,000 per year compared to those earning more than EUR 36,000 per year. Additionally, a relationship was found with birth weight, showing a 47.3% higher risk of overweight/obesity for each additional kilogram at birth, as well as with household difficulties in making ends meet.

## 4. Discussion

This study attempts to examine how home environment is related to child weight and health status, especially in the differences between high- and low-income households. In addition, our study aims to characterize the sample analyzed in the city of Valencia, to compare with other countries or cities. In this aspect, the descriptive data of the sample indicate high BMIs based on Z-scores, with 22.6% classified as overweight and 18.1% as obese, resulting in a total of 40.7% of participants having excess weight. The figures showed no significant differences between those who completed the survey and those who did not regarding obesity (18.7% and 17.6%, respectively). However, overweight prevalence was slightly higher among those who did not participate in the second part of the study (26.8%, compared to 19.8% among those who did). This could suggest a potential participation bias in households where children had higher overweight levels, although this was minimized by the sample size. Nevertheless, this should be considered in future research.

Other studies have reported similar data at the national level. For example, various ALADINO waves in Spain found a 40.6% prevalence of excess weight in 2019, with 23.3% classified as overweight and 17.3% as obese [5]. Regionally, similar trends have been reported. In 2002, de Bont et al. [17] observed prevalence of overweight and obesity of between 37.6% (95% CI: 37.3–38.0%) in 2016 and 41.9% (95% CI: 41.5–42.2%) in 2006, in a cohort of 1.1 million children and adolescents (aged 2 to 17 years). Obesity and overweight were more common in boys than in girls and in children of foreign nationalities, particularly from North and South America, compared to those of African, Asian, or European descent for both sexes. In this study, the highest prevalence of overweight and obesity was found for children between the ages of 6 and 11, with 44.9% in boys and 41.8% in girls in 2006, which decreased to 39.9% in boys and 37.6% in girls by 2016 [17]. In a cohort of 2.5 million children aged 2 to 17 years across seven Spanish autonomous communities, a 35.6% prevalence of excess weight (21.4% overweight and 14.2% obese) was reported between 2005 and 2017 [17]. The peak in overweight and obesity occurred again between ages 6 and 11, with 42.2% (95% CI: 41.8–42.5%) in 2005, decreasing to 37.8% (95% CI: 37.5–38.1%) in 2017 [18].

However, other national studies report lower figures. For example, among children aged 2 to 17 years, a study found a 28.6% prevalence rate of overweight or obesity [19] based on interviews conducted with 23,860 households, including 6106 children under the age of 15. Obesity was reported as 10.3% [19]. These figures, however, were self-reported (parents or guardians were asked directly about weight and height), which introduces a potential bias, while the previous studies included directly or indirectly obtained measurements taken from the study subjects. A similar issue arose with the National SARS-CoV-2 sero-epidemiology study in Spain (ENE-COVID), which in October 2023 produced a map of childhood overweight and obesity prevalence based on self-reported weight and height data. The prevalence for the province where our study was conducted (Valencia) was 34.7% (20.5% overweight and 14.2% obese), compared to a national average of 33.7% for boys and 26% for girls, with a total of 10,543 children participating across Spain, 718 from the Valencian Community, and 329 from the province of Valencia [20]. The PASOS study, assessing 2892 children aged 8 to 16 years, estimated a prevalence of 33.4% for overweight and obesity combined [7]. In other regional studies, cohorts followed from birth to age six reported prevalence rates of overweight and obesity of 31.9% in boys and 31.1% in girls, with a significant relationship between social vulnerabilities and higher rates of overweight and obesity from the age of two [21]. In a recently published study for a population between 6 and 9 years of age in the city of Valencia, with 2724 children recruited, a prevalence of 37.5% of excess weight was found [22].

Regarding dietary habits, data revealed concerning trends, with 86.7% of the sample not following dietary patterns compatible with the Mediterranean diet. These figures are similar to those reported in other studies using the same survey, such as ALADINO 2019, which found that only 12.1–15.2% of the sample adhered to an optimal Mediterranean diet, while 84.8–87.9% followed a poor-quality diet [5]. In this case, no significant differences were observed across different weight status categories, suggesting that poor diet is prevalent across all groups. This finding is consistent with a 2015 systematic review, which included 18 cross-sectional studies of individuals aged 2 to 25 years old (24,067 participants) and found a high adherence to the Mediterranean diet in only 10% (95% CI: 7–13%) of the sample [23]. While low adherence to the Mediterranean diet was not a significant explanatory factor for worse weight outcomes in this study, KidMed assessment has shown significant correlations in other countries. In Greece, a worse KidMed score was significantly associated with a higher prevalence of overweight and obesity, with an average score of 4.65 ± 2.14 among overweight or obese children, compared to 5.16 ± 2.05 in those of normal weight (*p*-value: 0.002) [24]. Similarly, in Turkey, children with lower KidMed scores also had higher rates of overweight and higher BMIs and waist or neck circumferences (*p*-value: <0.05), with poorer KidMed scores in obese children compared to those who were overweight (*p*-value: <0.05) [25]. Thus, assessing adherence to the Mediterranean diet remains critical, as it is recognized as a protective factor against excess weight in childhood.

Regarding adequate food access, the FIES scale is a relatively new tool, and its use has been prominent in recent years. The results of this study show that nearly half of the analyzed households (45.7%) experienced food insecurity, with most cases being mild (164 households, or 39.6% of the total). Notably, food insecurity was moderate or severe in 25 households (6% of the sample). These findings are consistent with national data. An analysis of 1990 households found that between 6.09% (95% CI: 5.21–6.98%) and 12.01% (95% CI: 9.65–14.37%) of children experienced moderate or severe food insecurity, with a global average of 41% of households experiencing moderate or severe food insecurity across 147 countries [26].

The relationship between food insecurity and overweight or obesity remains ambiguous. In a Spanish study of 1938 children aged 2 to 14, moderate-to-severe food insecurity was present in 7.7% of the sample (95% CI: 6.6–9.0), with higher rates of overweight (33.1%) and obesity (28.4%) in these households [26]. The relative risk (RR) for overweight in children from food-insecure households compared to food-secure households was 2.41 (95% CI: 1.5–4.0), and it was 1.99 (95% CI: 1.2–3.4) for obesity [27]. However, in higher-income countries like the United States, food insecurity has been linked to poorer physical health, greater exposure to smoking, and lower physical activity levels, but not necessarily to obesity [28]. In studies conducted on other continents, food insecurity has been significantly associated with poorer general health outcomes but not necessarily with higher rates of overweight or obesity, as seen in Ecuador [29], Brazil [30], and Mexico [31]. In Europe, in the Eastern region (Romania, Turkey, Bulgaria, and Lithuania), where better macro-economic indicators are related to lower rates of childhood overweight, comparing 5206 school children aged 6 to 11 years old the data indicates the proportion of overweight and obesity among school children is 18% and 6.5%, respectively; at the same time, the prevalence in Western Europe (Germany, Netherlands, and Italy) is 10.3% and 1.6%, respectively, for a similar sample [32]. These differences may be influenced by confounding factors such as infrastructure, geography, or cultural attitudes towards mobility and physical activity in different regions, with some Latin American studies even showing lower overweight rates in areas with higher food insecurity [29,33,34], or even higher rates of malnutrition [33,35]. In recent years, FIES has been increasingly recognized as an indicator of poor general health, with its relationship to overweight, obesity, or malnutrition depending on the economic capacity of the country in question. In higher-income countries, food insecurity tends to be associated with obesity, while in lower-income countries, it is linked to malnutrition [36,37].

Regarding income-related factors, it is noteworthy that respondents were often hesitant to disclose their household income. The data shows that one in three households in the total sample (137 households, or 33.1%) reported earning less than EUR 18,000 per year. In relation to the question of whether households were able to make ends meet, data from 401 households revealed that 56% had some difficulty making ends meet, with 12.3% reporting serious difficulties or an inability to do so. Statistical analysis found a significant relationship between gross annual household income and overweight/obesity risk, with a 3.6 times higher risk in children from households earning less than EUR 12,000 compared to those earning more than EUR 36,000. This finding is consistent with the scientific literature. A national study by García-Solano et al. [8] concluded that excess weight was more prevalent in families with lower socioeconomic status, particularly in those earning less than EUR 18,000 per year, where obesity prevalence reached 23.2%, double the rate in children from higher-income families. In households earning less than EUR 18,000, the combined prevalence of obesity and overweight was 47.3%, compared to 33.7% in those with incomes up to EUR 30,000, suggesting that higher income acted as a protective factor (RR 0.58, 95% CI: 0.47–0.71). A significant protective relationship was found between higher income and lower obesity rates when these variables were included in a multinomial logistic regression model (with younger age, lower parental BMI, or higher income being protective factors) [8]. Similarly, a study in Catalonia with 1.1 million participants over ten years found that obesity was more common in economically deprived areas (34,552 cases vs. 17,625), with no significant differences between urban and rural areas [17]. In this cohort study, covering all ages (2 to 17 years), both sexes, and the years 2006 to 2016, living in a poor area was a significant risk factor (RR 1.07, 95% CI: 1.00–1.15 to RR 2.24, 95% CI: 1.98–2.54), with a statistically significant correlation in all cases (*p* < 0.001) [17].

Other studies examining socioeconomic levels vary in how they collect this data but reach similar conclusions. For example, the PASOS 2022 study [7] stratified data by the average income per person in the census section where the school was located. Its conclusions indicated that individuals from lower-income households had higher rates of overweight and obesity (Q1 or <EUR 10,317 annually, 38% overweight and obesity vs. Q4 or >EUR 14,335 annually, 29.3% prevalence) [7]. Iguacel et al. [21] also found that as social vulnerabilities increased, so did the prevalence of overweight and obesity, with a 26% prevalence in children with no vulnerabilities, rising to 50.5% in those with three or more vulnerabilities. These vulnerabilities were defined as belonging to a minority group, parental educational level, or parental employment status, used as a proxy for household economic status [21].

In other regions this is also the case, although not always to a significant extent. For example, a study in Brazil shows that the population with obesity in households with incomes below BRL 19,620 was 29% (n = 20), while in households where income tripled to BRL 58,860, the prevalence was 20.3% (n = 14), but the relationship was not significant [38]. On the other hand, a study in Taiwan showed a significant relationship between obesity or overweight prevalence in non-low-income households (22.4%) and low-income households (30.5%), with an adjusted odds ratio of 1.44 (95% CI: 1.29–1.60) [39]. In our study, we also observed an average increase of 2.0 kg (95% CI: −0.2–4.3) in students from households with some difficulty making ends meet, and 4.8 kg (95% CI: 1.3–8.3) in those from households reporting serious difficulties or an inability to make ends meet. It seems appropriate that economic capacity and the ability to make ends meet were self-reported rather than inferred from census data, which increased the sensitivity of the analysis. However, regardless of how economic-related social determinants are measured, all cases found a higher risk of obesity or overweight associated with poorer resources, inversely proportional to income.

### Strengths and Limitations

Among the strengths of the study is that the data aligns with reference studies for monitoring obesity and overweight, and that the information was collected objectively, particularly in relation to nutritional information being collected through a structured telephone survey, rather than self-administered surveys. This approach reduces potential biases, such as information bias or social desirability bias, because in health studies, especially in children’s health studies, there is a tendency to overestimate health or diet quality and underestimate risk factors.

However, this also represents a potential limitation, as the requirement for the follow-up survey to be interviewer-led reduced family participation due to the length of the survey, as noted in the methodology and results sections. Additionally, this method is not free from recall bias. Another limitation is the cross-sectional design, which precludes the establishment of causal relationships, and the fact that the analysis is limited to a single city. Furthermore, certain groups in the recruited sample were underrepresented, which prevented a more in-depth analysis of some covariates, such as the results from the FIES scale.

## 5. Conclusions

At least 40.7% of the analyzed sample of children aged 5 to 14 are affected by excess weight, while nearly half of the households experience mild food insecurity, with 6% facing moderate or severe levels. This prevalence is influenced by health disparities linked to economic capacity, particularly household difficulties in making ends meet and annual income levels. The child population does not exhibit good dietary habits compatible with the Mediterranean diet across any social stratum, which could lead to health problems in adulthood.

It is essential to continue promoting epidemiological surveillance of these conditions and their relationship with social determinants with further studies involving larger samples that encompass cities and neighborhoods with significant socioeconomic differences, to establish and implement action towards greater health equity.

Institutional efforts and policy decisions to address overweight and obesity in this population or their households should incorporate information on health determinants linked to purchasing power to develop appropriate health programs for the most vulnerable population groups.

## Figures and Tables

**Table 1 nutrients-17-02006-t001:** Anthropometric parameters of the total sample and by sex, along with corresponding assessments for those whose parents provided responses regarding social determinants.

Variable		Total Sample (n: 698)	Boys (n: 360)	Girls (n: 338)	Total Sample with Social Determinants (n: 414)	Boys (n: 207)	Girls (n: 207)
**Age (years)**	**Mean (SD)**	9.80 (2.90)	9.84 (2.96)	9.83 (2.81)	10.10 (3.00)	10.20 (2.98)	10.04 (2.96)
	**Median (P_25_–P_75_)**	10 (7–13)	10 (7–13)	10 (7–12)	10 (8–13)	11 (8–13)	10 (7–13)
**Weight (kg)**	**Mean (SD)**	40.70 (15.70)	40.92 (16.45)	40.48 (14.85)	41.80 (15.50)	42.7 (16.0)	40.8 (15.0)
	**Median (P_25_–P_75_)**	39.00 (28.50–50.70)	38.8 (27.85–51.17)	39.25 (29.54–48.9)	40.30 (30.10–52.30)	41.00 (30.30–53.60)	39.44 (29.10–51.94)
**Height (cm)**	**Mean (SD)**	143.00 (17.30)	143.05 (18.76)	142.45 (15.62)	144.55 (17.30)	145.89 (18.12)	142.97 (16.46)
	**Median (P_25_–P_75_)**	144.00 (129.00–156.00)	141.3 (128.00–157.52)	143.85 (129.8–155.47)	145.40 (131–158)	145.6 (132.10–160.20)	145.5 (129.30–157.40)
**BMI (kg/m^2^)**	**Mean (SD)**	19.30 (4.14)	19.21 (4.12)	19.31 (4.17)	19.30 (4.01)	19.40 (3.98)	19.28 (4.04)
	**Median (P_25_–P_75_)**	18.40 (16.20–21.40)	18.23 (16.17–21.38)	18.59 (16.22–21.46)	18.30 (16.35–21.46)	18.44 (16.43–21.86)	18.34 (16.34–21.38)
**Z-Score**	**Mean (SD)**	0.68 (1.36)	0.73 (1.41)	0.61 (1.31)	0.65 (1.30)	0.75 (1.35)	0.56 (1.25)
	**Median (P_25_–P_75_)**	0.62 (−0.31–1.59)	0.61 (−0.34–1.76)	0.62 (−0.27–1.48)	0.57 (−0.31–1.53)	0.59 (−0.32–1.87)	0.47 (−0.29–1.34)
	**Undernutrition (%)**	16 (2.30%)	8 (2.22%)	8 (2.36%)	6 (1.40%)	2 (1.00%)	4 (1.90%)
	**Normal weight (%)**	398 (57.00%)	201 (55.73%)	197 (58.28%)	253 (61.10%)	123 (59.40%)	130 (62.80%)
	**Overweight (%)**	158 (22.60%)	75 (20.83%)	83 (24.56%)	82 (19.80%)	38 (18.40%)	44 (21.30%)
	**Obese (%)**	126 (18.10%)	76 (21.11%)	50 (14.80%)	73 (17.60%)	44 (21.30%)	29 (14.00%)
**Waist (cm)**	**Mean (SD)**	64.90 (10.70)	65.97 (10.55)	63.74 (10.77)	65.2 (10.2)	66.8 (9.8)	63.5 (10.30)
	**Median (P_25_–P_75_)**	64.00 (57.50–71.00)	64.45 (58.30–72.50)	63.00 (57.00–68.95)	64.00 (58.10–71.00)	65.00 (60.00–74.00)	62.00 (58.50–69.00)
**Hip (cm)**	**Mean (SD)**	79.70 (12.90)	79.04 (13.00)	80.31 (12.74)	80.5 (12.7)	80.5 (12.4)	80.6 (13.0)
	**Median (P_25_–P_75_)**	80.00 (69.50–88.40)	79.65 (69.00–87.52)	80.00 (70.00–89.00)	81.0 (70.2–89.2)	81 (71–89)	81 (70–91)
**Waist–Hip Index**	**Mean (SD)**	0.818 (0.078)	0.83 (0.07)	0.79 (0.08)	0.813 (0.075)	0.833 (0.065)	0.793 (0.080)
	**Median (P_25_–P_75_)**	0.82 (0.77–0.86)	0.83 (0.79–0.87)	0.80 (0.75–0.84)	0.81 (0.77–0.86)	0.83 (0.79–0.87)	0.79 (0.74–0.84)
**Waist–Height Index**	**Mean (SD)**	0.45 (0.06)	0.46 (0.06)	0.45 (0.06)	0.45 (0.06)	0.46 (0.06)	0.44 (0.06)
	**Median (P_25_–P_75_)**	0.45 (0.42–0.49)	0.45 (0.42–0.50)	0.44 (0.41–0.48)	0.45 (0.41–0.48)	0.45 (0.41–0.49)	0.44 (0.41–0.47)
**BC (cm)**	**Mean (SD)**	22.60 (4.70)	22.66 (5.18)	22.44 (4.12)	22.60 (4.20)	22.90 (4.30)	22.30 (4.06)
	**Median (P_25_–P_75_)**	22.00 (19.30–25.00)	22.00 (19.00–25.43)	22.00 (19.5–24.95)	22.00 (19.50–25.00)	22.30 (19.60–25.80)	21.90 (19.50–24.70)
**TS (mm)**	**Mean (SD)**	15.70 (8.30)	14.54 (8.17)	16.97 (8.30)	15.60 (8.200)	14.30 (7.99)	16.90 (8.24)
	**Median (P_25_–P_75_)**	14.00 (10.00–20.00)	12.00 (8.00–16.25)	15.00 (11.00–21.00)	13.00 (10.00–20.00)	12.00 (8.00–20.00)	15.00 (11.00–21.00)
**FA (kg)**	**Mean (SD)**	17.10 (8.30)	17.97 (9.47)	16.25 (6.75)	17.90 (8.50)	19.00 (9.54)	16.70 (7.05)
	**Median (P_25_–P_75_)**	15.30 (10.80–22.50)	15.20 (10.90–24.20)	15.40 (10.70–21.30)	16.60 (11.00–23.40)	17.0 (11.30–24.70)	16.10 (10.60–22.40)

SD—Standard Deviation. P—Percentile. BMI—Body Mass Index. FA—Grip Strength. BC—Brachial Circumference. TS—Tricipital Skinfold.

**Table 2 nutrients-17-02006-t002:** Personal characteristics, dietary habits, physical activity and screen time of the sample for which household responses were received (n: 414).

Variable			Households (n: 414)	Boys				Girls			
				Total	Normal Weight	Overweight	Obese	Total	Normal Weight	Overweight	Obese
				(n: 207)	(n: 123)	(n: 38)	(n: 44)	(n: 207)	(n: 130)	(n: 44)	(n: 29)
**Birth weight (kg)**	**Mean (SD)**		3.18 (0.64)	3.01 (0.64)	2.93 (0.63)	3.18 (0.77)	3.11 (0.52)	2.95 (0.63)	2.88 (0.62)	3.02 (0.59)	3.11 (0.73)
	**Median (P_25_–P_75_)**		3.20 (2.83–3.60)	3.30 (2.90–3.60)	3.22 (2.85–3.60)	3.47 (3.00–3.76)	3.3 (3–3.65)	3.15 (2.85–3.60)	3.22 (2.7–3.45)	3.20 (2.82–3.50)	3.2 (2.75–3.76)
**Exclusive breastfeeding**	**No answer or <6 months (%)**		250 (60.40%)	132 *	70	30	30	118 *	77	21	16
	**≥6 months (%)**		164 (39.6%)	75	53	8	14	89	53	23	13
**Breastfeeding (duration)** **Mean (SD) duration by weight status, months**	**Unknown**		6 (1.70%)	12.61 (15.61)	12.96 (14.02)	7.97 (11.51)	16.11 (21.28)	12.91 (14.13)	14.47 (14.77)	12.23 (16.05)	7.27 (7.17)
	**0 or <1 month**		83 (20.00%)								
	**1 to 5 months**		70 (17.10%)								
	**6 to 11 months**		80 (19.60%)								
	**12 to 23 months**		86 (21.00%)								
	**24 months or more**		89 (21.80%)								
**Dietary quality (KidMed) and FIES**											
**Total Score**	**Mean (SD)**		5.08 (2.18)	4.97 (2.11)	4.93 (2.12)	4.84 (2.06)	5.2 (2.19)	4.79 (2.24)	4.6 (2.16)	5.27 (2.47)	5 (2.23)
	**Median (P_25_–P_75_)**		5 (4–7)	5 (4–7)	5 (4–7)	5 (4–6)	6 (4–6)	5 (4–7)	5 (3–6)	6 (5–7)	5 (3–7)
**Very low quality diet (score ≤ 3)**			115 (27.80%)	53 *	33	9	10	62 *	41	9	9
**Need to improve dietary patterns (score 4 to 7)**			244 (58.90%)	128	76	24	27	116	73	26	16
**Optimal Mediterranean diet (score ≥ 8)**			55 (13.30%)	26	14	5	7	29	16	9	4
**No food insecurity**			225 (54.30%)	113	66	19	27	112	71	24	12
**Mild food insecurity**			164 (39.60%)	82	54	17	10	82	45	18	17
**Moderate food insecurity**			19 (4.60%)	9	2	2	5	10	8	2	0
**Severe food insecurity**			6 (1.40%)	3	1	0	2	3	3	0	0
**Screen Time Behavior**											
**Hours per day spent watching television or using a computer, video games, tablets, or phone.**	**Weekdays**	**Mean (SD)**	1.78 (1.23)	1.68 (0.99)	1.60 (0.97)	1.78 (0.95)	1.77 (1.06)	1.87 (1.42)	1.91 (1.47)	1.56 (0.83)	2.13 (1.84)
		**Median (P_25_–P_75_)**	2 (1–2)	2.00 (1.00–2.00)	1.83 (1.00–2.00)	2.00 (1.00–2.00)	2.00 (1.00–2.00)	2.00 (1.00–2.00)	1.50 (1.00–2.00)	1.50 (1.00–2.00)	2.00 (1.00–2.00)
	**Weekends**	**Mean (SD)**	2.96 (1.56)	2.88 (1.40)	2.76 (1.28)	3.42 (1.15)	2.75 (1.57)	3.04 (1.70)	3.11 (1.71)	2.69 (1.29)	3.23 (2.21)
		**Median (P_25_–P_75_)**	3 (2–4)	3.00 (2.00–4.00)	3.00 (2.00–2.57)	4.00 (3.00–4.00)	2.75 (2.00–4.00)	3.00 (2.00–4.00)	3.00 (2.00–4.00)	3.00 (2.00–4.00)	3.00 (2.00–4.00)
**Physical Activity**											
**No sport outside the classroom**			125 (30.2%)	-	-	-	-	-	-	-	-
**1–3 h/week**			163 (39.4%)	-	-	-	-	-	-	-	-
**≥4 h/week**			126 (30.4%)	-	-	-	-	-	-	-	-
**Weight Status by Nationality (Birth)**											
**Sample**			**Total**	**Undernutrition**		**Normal weight**		**Overweight**		**Obese**	
**Spanish**			304	5 (1.6%)		188 (61.8%)		57 (18.8%)		54 (17.8%)	
**Foreign**			110	1 (1.0%)		62 (59%)		24 (22.9%)		18 (17.1%)	

* Includes cases where weight assessment was consistent with undernutrition, but the sample in that category (n: 6) is included under ‘total’.

**Table 3 nutrients-17-02006-t003:** Anthropometric parameters of the sample according to their purchasing power and their difficulties in making ends meet.

Variable		Annual Income Groups (n: 322)				*p*-Value	Reported Difficulty in Making Ends Meet Groups (n: 401)			*p*-Value
		<12,000 €	12,000–18,000 €	18,001–36 K€	>36,000 €		Not Making It or Problems	Some Problems	Any Problems	
Sample		n: 65	n: 72	n: 113	n: 72		n: 51	n: 168	n: 182	
**Weight (kg)**	**Mean (SD)**	44.50 (16.10)	43.40 (15.30)	42.20 (16.10)	37.20 (12.00)	0.022	47.10 (16.40)	42.30 (15.60)	39.90 (15.00)	0.013
	**Median (P_25_–P_75_)**	42.60 (31.90–54.20)	42.20 (31.10–53.20)	40.90 (30.20–54.60)	37.10 (26.80–44.50)		45.10 (34.80–55.00)	41.00 (30.50–53.00)	38.4 (26.80–50.70)	
**BMI (kg/m^2^)**	**Mean (SD)**	20.52 (4.33)	19.80 (4.46)	19.17 (3.85)	17.69 (2.45)	<0.001	21.37 (4.53)	19.53 (4.17)	18.57 (3.51)	<0.001
	**Median (P_25_–P_75_)**	19.90 (17.20–23.10)	18.50 (16.70–21.40)	18.10 (16.20–21.80)	17.30 (15.80–19.40)		21.00 (18.20–23.10)	18.30 (16.50–21.60)	17.70 (15.80–20.40)	
**Z-Score**	**Mean (SD)**	1.08 (1.51)	0.67 (1.38)	0.61 (1.22)	0.24 (1.10)	0.003	1.16 (1.43)	0.71 (1.35)	0.43 (1.18)	0.001
	**Median (P_25_–P_75_)**	1.04 (−0.01–2.30)	0.45 (−0.33–1.61)	0.48 (−0.31–1.41)	0.11 (−0.51–0.93)		1.26 (0.18–2.30)	0.60 (−0.29–1.58)	0.33 (−0.47–1.21)	
	**Undernutrition (%)**	2 (3.10%)	2 (2.80%)	1 (0.90%)	0 (0.00%)		2 (3.90%)	3 (1.80%)	1 (0.50%)	
	**Normal weight (%)**	30 (46.20%)	43 (59.70%)	70 (61.90%)	55 (76.40%)		20 (39.20%)	104 (61.90%)	122 (67.00%)	
	**Overweight (%)**	14 (21.50%)	14 (19.40%)	23 (20.40%)	11 (15.30%)		14 (27.50%)	28 (16.70%)	38 (20.90%)	
	**Obese (%)**	19 (29.20%)	13 (18.10%)	19 (16.80%)	6 (8.30%)		15 (29.40%)	33 (19.60%)	21 (11.50%)	
**Waist (cm)**	**Mean (SD)**	67.80 (11.10)	66.50 (10.5)	65.20 (9.60)	61.60 (8.10)	0.002	67.90 (12.70)	66.20 (10.5)	63.50 (9.00)	0.008
	**Median (P_25_–P_75_)**	66.0 (59.00–74.00)	65.40 (58.60–71.40)	63.60 (58.20–71.00)	61.20 (56.80–66.20)		68 (59–75)	64 (59–72)	64 (57–69)	
**Hip (cm)**	**Mean (SD)**	83.40 (12.50)	82.70 (12.40)	80.50 (13.30)	76.50 (10.30)	0.004	85.10 (12.90)	81.20 (12.40)	78.80 (12.70)	0.006
	**Median (P_25_–P_75_)**	82.00 (77.00–91.00)	83.00 (73.70–90.70)	81.50 (70.00–90.50)	76.80 (67.90–85.00)		86 (77–92)	81 (72–90)	79 (68–88)	
**Waist–Hip Index**	**Mean (SD)**	0.814 (0.063)	0.806 (0.065)	0.815 (0.074)	0.811 (0.088)	0.858	0.798 (0.097)	0.818 (0.066)	0.813 (0.077)	0.258
	**Median (P_25_–P_75_)**	0.81 (0.77–0.87)	0.80 (0.77–0.84)	0.82 (0.75–0.86)	0.82 (0.77–0.86)		0.80 (0.75–0.87)	0.82 (0.77–0.86)	0.81 (0.76–0.85)	
**Waist–Height Index**	**Mean (SD)**	0.469 (0.070)	0.457 (0.065)	0.450 (0.051)	0.434 (0.051)	0.006	0.466 (0.081)	0.459 (0.063)	0.443 (0.049)	0.014
	**Median (P_25_–P_75_)**	0.46 (0.41–0.53)	0.44 (0.41–0.50)	0.45 (0.41–0.48)	0.43 (0.40–0.46)		0.47 (0.41–0.52)	0.45 (0.42–0.49)	0.44 (0.41–0.47)	
**BC (cm)**	**Mean (SD)**	23.60 (4.32)	23.1 (4.14)	22.4 (4.23)	21.2 (3.14)	0.001	24.30 (4.24)	22.80 (4.30)	22.00 (4.00)	0.002
	**Median (P_25_–P_75_)**	23.30 (21.30–25.90)	22.30 (20.40–25.20)	21.50 (19.00–25.90)	21.00 (18.90–22.90)		24 (22–28)	22 (20–25)	21 (19–25)	
**TS (mm)**	**Mean (SD)**	17.30 (9.33)	16.90 (9.76)	15.00 (7.83)	12.70 (5.73)	0.001	19.10 (9.48)	15.90 (8.65)	14.20 (7.10)	0.002
	**Median (P_25_–P_75_)**	16.00 (9.00–22.00)	14.00 (10.00–23.00)	13.00 (9.00–20.00)	11.00 (9.00–16.00)		18 (12–25)	13 (10–20)	12 (9–19)	
**FA (kg)**	**Mean (SD)**	18.10 (8.83)	18.50 (7.71)	18.30 (8.89)	16.40 (7.63)	0.405	20.00 (8.90)	17.90 (8.47)	17.40 (8.15)	0.135
	**Median (P_25_–P_75_)**	15.60 (11.20–24.40)	18.20 (11.90–23.70)	16.70 (11.10–24.10)	15.60 (10.10–21.40)		20.00 (12.10–26.30)	16.50 (10.80–23.90)	16.40 (11.00–22.50)	

SD—Standard Deviation. P—Percentile. BMI—Body Mass Index. FA—Grip Strength. BC—Brachial Circumference. TS—Tricipital Skinfold.

**Table 4 nutrients-17-02006-t004:** Multivariate models (95% CI) for student weight and Z-score according to different health determinants.

Health Determinants		Multivariate Models for Student Weights (n: 329)				Multivariate Models for Z-Score (n: 329)			
		Beta	95% CI		*p*-Value	Beta	95% CI		*p*-Value
**Birth Weight (n: 388)**		2628	0.981	4.274	0.002	0.304	0.095	0.514	0.005
**Annual household income (n: 322)**	**>EUR 36,000 (n: 72)**	-			-	(baseline)			0.009
	**>EUR 18,000** **–** **36,000 (n: 113)**	-	-	-	-	1.941	0.998	3.774	0.051
	**>EUR 12,000** **–** **18,000 (n: 72)**	-	-	-	-	2.031	0.983	4.200	0.056
	**≤** **EUR 12,000 (n: 65)**	-	-	-	-	3.559	1.706	7.422	0.001
**Difficulty in making ends meet (n: 401)**	**Any problems (n: 182)**	(baseline)			0.016	(baseline)			<0.001
	**Some problems (n: 168)**	2.023	−0.224	4.271	0.078	0.375	0.089	0.661	0.01
	**Not making it, or problems (n: 51)**	4.819	1.348	8.290	0.007	0.906	0.464	1.348	<0.001

## Data Availability

The original contributions presented in this study are included in the article/Appendix A. Further inquiries can be directed to the corresponding author.

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
