# Peer review of "Impact of Social and Economic Determinants on the Prevalence of Childhood Overweight and Obesity: A Cross-Sectional Study from the ENPIV in Valencia, Spain"

_nutrients, 2025, doi:10.3390/nu17122006_

Round 1

Reviewer 1 Report

Comments and Suggestions for Authors

I recommend using STROBE to support the report.

Vulnerability is a key issue in the manuscript, which deserves to be further discussed in the discussion. What is the profile and lifestyle of these children who are part of less privileged families (≤ E 36,000) compared to children from more privileged families?

Also, how can actions, strategies and public policies reverse this situation of inequality? Are there any current actions in the city?

Author Response

The authors would like to sincerely thank the reviewers for their valuable efforts in helping improve our work. Please find attached a document in which we respond to your comments, as well as those from the other two reviewers, to provide a comprehensive overview of the changes made in the revised version of the manuscript.

We will upload both the manuscript and the supplementary material (including the STROBE checklist) in two formats: a "track" version, with changes highlighted in red, and a "clean" version, with all changes unmarked. We hope we have adequately addressed all your concerns, which have significantly contributed to strengthening our manuscript.

Reviewer 2 Report

Comments and Suggestions for Authors

Dear Authors,

Your study titled, “Impact of Social and Economic Determinants on the Prevalence of Childhood Overweight and Obesity: A Cross-Sectional Study from the ENPIV in Valencia, Spain”, was well-written and adds nicely to the body of literature.  With the high incidence of childhood obesity, it is crucial to identify reasons and possible changes to effectively reduce obesity in this population.

Below are minor comments that may improve the readers understanding:

  1. Abstract: The methods are not clear. What tools were used to collect the data?
  2. Introduction: Line 47- should it say Spain is the fourth country in terms of having the highest percentage of childhood overweight and obesity?
  3. Methods: Line 140: Consider stating the full name for ISAK (International Society for the Advancement of Kinanthropometry) for the readers that are not familiar with the method.
  4. Tables: Consider aligning the columns to the left instead of center to make the table easier to read. If possible, consider making the font size slightly smaller to improve the fit of the text in the table.  Also, Table 3 is easier to read as the SD is below the Mean versus next to it.  This does make a table longer but does improve readability.  Just something to consider- not a necessary revision if you do not agree.

Author Response

(The authors gave the same response as above.)

Reviewer 3 Report

Comments and Suggestions for Authors

This is a carefully performed analysis of the social determinants of obesity in a Spanish city.  The analysis is well done and described well.  The conclusions are well supported.

Author Response

(The authors gave the same response as above.)

Round 2

Reviewer 1 Report

Comments and Suggestions for Authors

The changes made improved the report. Congratulations.